# Distinct microbial nitrogen cycling processes in the deepest part of the ocean

Yuhan Huang,[1,2,3] Xinxu Zhang,[1,2] Yu Xin,[4] Jiwei Tian,[5] Meng Li[1,2]

**ABSTRACT** The Mariana Trench (MT) is the deepest part of the ocean on Earth. Previous studies have described the microbial community structures and functional potential in the seawater and surface sediment of MT. Still, the metabolic features and adaptation strategies of the microorganisms involved in nitrogen cycling processes are poorly understood. In this study, comparative metagenomic approaches were used to study microbial nitrogen cycling in three MT habitats, including hadal seawater [9,600–10,500 m below sea level (mbsl)], surface sediments [0–46 cm below seafloor (cmbsf) at a water depth between 7,143 and 8,638 mbsl], and deep sediments (200–306 cmbsf at a water depth of 8,300 mbsl). We identified five new nitrite-oxidizing bacteria (NOB) lineages that had adapted to the oligotrophic MT slope sediment, *via* their $CO_2$ fixation capability through the reductive tricarboxylic acid (rTCA) or Calvin-Benson-Bassham (CBB) cycle; an anammox bacterium might perform aerobic respiration and utilize sedimentary carbohydrates for energy generation because it contains genes encoding type A cytochrome *c* oxidase and complete glycolysis pathway. In seawater, abundant alkane-oxidizing *Ketobacter* species can fix inert $N_2$ released from other denitrifying and/or anammox bacteria. This study further expands our understanding of microbial life in the largely unexplored deepest part of the ocean.

**IMPORTANCE** The metabolic features and adaptation strategies of the nitrogen cycling microorganisms in the deepest part of the ocean are largely unknown. This study revealed that anammox bacteria might perform aerobic respiration in response to nutrient limitation or $O_2$ fluctuations in the Mariana Trench sediments. Meanwhile, an abundant alkane-oxidizing *Ketobacter* species could fix $N_2$ in hadal seawater. This study provides new insights into the roles of hadal microorganisms in global nitrogen biogeochemical cycles. It substantially expands our understanding of the microbial life in the largely unexplored deepest part of the ocean.

**KEYWORDS** anammox, deep biosphere, hadal trench, metagenomics, nitrite oxidation

The hadal biosphere refers to a trench area with a water depth greater than 6,000 m that accounts for 45% of the vertical depth of the ocean. This habitat is characterized by high hydrostatic pressure (>60 MPa), low temperature (<4°C), and relatively isolated environments (1–3). In general, the topography of a trench is shaped like a funnel consisting of slope and axis area. Importantly, the high rates of carbon turnover were measured in the sediments, which contributes to the accumulation of organic carbon and nitrogen compounds (e.g., nitrate and amino acids) (4–6). However, compared with the trench axis, the trench slope is steeper with a more complex topography (e.g., horst, graben, seamount, and ridges) and covers a much wider area (7, 8). Sinking organic material unevenly distributed on the slope can also slide from the top to the trench bottom *via* ocean crust movements (such as earthquakes) and seawater circulation.

Address correspondence to Meng Li, limeng848@szu.edu.cn.

Yuhan Huang and Xinxu Zhang contributed equally to this article. Author order was determined by drawing straws.

The authors declare no conflict of interest.

See the funding table on p. 12.

Therefore, the sediment layer of the slope is generally thinner than that at the trench bottom and thus has a lower content of organics and microorganisms (4, 6, 9).

Previous studies have shown that various bacterial and archaeal communities were detected in the MT. For example, *Proteobacteria*, *Bacteroidota*, *Marinisomatota,* and *Thaumarchaeota* were dominant groups in the hadal seawater (10). The common microorganisms in the surface sediment were *Proteobacteria*, *Chloroflexota*, *Planctomycetes*, *Thaumarchaeota,* and *Nanoarchaeota* (9, 11). Recently, the metabolic features and adaptation strategies of the hadal microbiome have been uncovered using multiple techniques, such as genomics (metagenomics and metatranscriptomics), *in situ* measurements, and laboratory incubations. For example, *Thaumarchaeota* in MT seawater possessed two distinct sets of adenosine-triphosphate (ATP) synthases (binding $Na^+$ or $H^+$) in response to the extreme hadal environment (12). *Chloroflexi* in the sediments of MT can utilize a wide range of organics, and these bacteria respond to poor nutrition and a periodically fluctuating environment with a "feast-or-famine" metabolic strategy (13). *Labrenzia aggregate* strains isolated from the MT (0–9,600 m) had increased the plasmidic gene numbers along with isolation depth and obtained the extra genetic potential mainly through plasmid exchange to resist the extremely high hydrostatic pressure (14). Dermacozines, a new secondary metabolite produced by the bacterial strain *Dermacoccus abyssi* MT1.1$^T$ (isolated from MT sediment at 10,898 m), has multiple activities such as scavenging free radicals and cytotoxicity (15). Some *Alphaproteobacteria* isolates (e.g., *Sagittula stellata*, *Labrenzia aggregate*, *Pelagibaca bermudensis*) produce dimethylsulfoniopropionate, which protects bacteria against high hydrostatic pressure in the aphotic deep ocean (16). Taken together, adapting microbial communities to the hadal environment may contribute to their metabolic activities and important roles in the global carbon and nitrogen biogeochemical cycles.

Nitrogen is a fundamental nutrient element in the environment. The microbial nitrogen cycle is a basic metabolic process to synthesize biological molecules (e.g., amino acids, proteins, and nucleic acids) during microbial growth and reproduction. However, bioavailable nitrogen (e.g., nitrate and ammonia) is scarce on Earth, and the microbial-mediated nitrogen cycling processes contribute to a significant fraction of bioavailable nitrogen in many environments (17, 18). In general, the nitrogen compounds in hadal environments consist of $NO_3^-$, $NO_2^-$, $NH_4^+$, and proteins. $NH_4^+$ can be released through microbial processes such as deamination and protein degradation, which are catalyzed by ammonia lyases, deaminases, and proteases (19). Constant concentrations of $NO_3^-$ (34–36 µM) and $NH_4^+$ (<1 µM) were observed throughout the hadal water of MT (10, 20). However, concentrations of these nitrogen compounds varied with depth in the sediment of MT. Relatively high concentrations of $NO_3^-$ (20–36 µM) and low concentrations of $NO_2^-$ (~0.1 µM) and $NH_4^+$ (0–5 µM) were detected in the surface sediment (<50 cmbsf) (6), but these nitrogen compounds showed the opposite trends in the deep sediment (21). Few studies have reported microbial nitrogen metabolisms in the MT surface sediment and seawater. For example, multiple microbial groups (e.g., *Proteobacteria*, *Planctomycetota,* and *Bacteroidota*) participate in $NO_3^-$ reduction [dissimilatory nitrate reduction to ammonium (DNRA) and denitrification] (6, 22, 23). In the surface sediment of the Atacama Trench (5–10 cm at 8,085 m) and Kermadec Trench (15–20 cm at 10,010 m), approximately 67% and 91% of $N_2$ were derived from the anammox process (24). Furthermore, some anammox bacterial MAGs (metagenome-assembled genomes) were retrieved from the MT surface sediments (18–21 cm at 10,840 m) (6). The nitrogen fixation gene *nifH* was detected in MT seawater (>9,600 m) (25). For nitrification, ammonia-oxidizing archaea (AOA) dominated by *Thaumarchaeota* in hadal trenches perform autotrophic carbon fixation through ammonium oxidation (12). However, the metabolic features and adaption strategies of these anammox bacteria, other types of nitrifying bacteria (e.g., NOB), and $N_2$ fixers have not yet been revealed in the hadal biosphere.

To fill these gaps in our knowledge, we have used comparative metagenomic approaches to reveal the unique features of the microbial nitrogen cycling processes

in three different hadal habitats, including hadal seawater (9,600–10,500 mbsl), surface sediment (0–46 cmbsf at a water depth between 7,143 and 8,638 mbsl), and deep sediment (200–306 cmbsf at a water depth of 8,300 mbsl) (Table S1).

## MATERIALS AND METHODS

### Sample collection, DNA extraction, and metagenome sequencing

The nine deep sediment samples were collected from the MT slope in the western Pacific Ocean during a cruise in 2019 using the research vessel *Haida* (11.13° N, 142.33° E). The water depth was 8,300 mbsl (Fig. S1). One sediment core was collected by gravity corer, and the subsamples of 200, 213, 229, 242, 250, 260, 272, 290, and 306 cmbsf were used in this study. The samples were stored at −80℃ immediately after recovery onboard. DNA was extracted from 0.5 g of each sediment using a DNeasy PowerSoil Pro Kit (Qiagen, Germany) according to the manufacturer's instructions, and only the interior sections of the sediment were subsampled with a flame-sterilized scoop for DNA extraction. In parallel, blank controls for all sampling and DNA extractions were prepared using Milli-Q water (18.2 MΩ; Millipore, USA) filtered through the 0.22-µm mesh membrane. Metagenomic sequencing was performed on the NovaSeq 6000 platform (Illumina, USA) using 2 × 150 bp paired-end technology.

The six surface sediment samples were obtained from the MT slope at water depths between 7,143 and 8,638 mbsl as described elsewhere (6). All intact core samples were immediately subsampled onboard and frozen at −80℃ until use. DNA was extracted from 10 g sediment of each sample using PowerMax Soil DNA Isolation Kit (MoBio, Germany) according to the manufacturer's protocol. Sequencing was performed on NovaSeq 6000 platform (2 × 150 bp for S.7143.2 and S.7143.16) or Miseq platform (2 × 300 bp for S.7850.0, S.8638.0, S.8638.36, and S.8638.44) (Illumina, USA).

The three seawater samples were collected from the MT Challenger Deep at water depths between 9,600 and 10,500 mbsl as described elsewhere (26). About 50L of each seawater was filtered onto a 0.22-µm polycarbonate membrane (GTTP, 142 mm, Millipore, USA). DNA was extracted using the SDS method as described previously (26). Sequencing was performed on the Illumina HiSeq X-Ten platform using 2 × 150 bp paired-end technology.

### Metagenomic assembly and binning

All the metagenomes used in this study were assembled and binned using the same method. Briefly, the raw metagenomic reads were trimmed using Trimmomatic (v.0.38) (27) with default parameters. All clean reads from the six surface sediments were pooled together before *de novo* assembly to one coassembly. Meanwhile, all clean reads from the three seawater samples were pooled together before *de novo* assembly to one coassembly. The clean reads were then assembled into contigs using SPAdes (v.3.15.0) (28) with the parameters: --meta -k 21,29,39,59,79,99. Metagenomic binning and quality assessment were performed as described elsewhere (29). Briefly, the binning was performed using MetaBAT 2 (v.2.12.1) (30), and optimized with DAS Tool (v.1.1) (31). The completeness, contamination, and heterogeneity of the recovered MAGs were determined based on lineage-specific conserved marker gene sets in each genome by CheckM (v.1.0.7) (32). In total, 498 MAGs with >50% completeness and <10% contamination were selected for downstream analysis as described elsewhere (6, 13, 22).

### Taxonomic classification and relative abundance

Taxonomic assignments of the MAGs were performed using the "classify_wf" workflow in the GTDB-Tk software (v.2.1.0) (33). The recovered MAGs were dereplicated at 95% identity with CoverM software (v.0.6.1) (https://github.com/wwood/CoverM) to avoid arbitrary mapping among highly similar genomes, and the relative abundance of the dereplicated MAGs in each metagenome was calculated using CoverM with the program

"coverm genome." The total microbial community composition of each metagenome was performed using Metaxa2 (v.2.1.3) (34) based on the identified 16S rRNA gene reads. The RPKM value {RPKM = Mapped reads/[Gene length (Kb) ×Total reads (million)]} was used to indicate the abundance of each nitrogen cycling gene, which was normalized to account for variations in gene length and data set size (3). Statistical significance of the relative abundance between the two samples was analyzed using the Mann-Whitney U test in SPSS (v.22.0), and differences were considered significant when $P < 0.05$ as described elsewhere (35, 36).

## Retrieval of anammox, NOB, and *Ketobacter* reference genomes

The collection of anammox, NOB, and *Ketobacter* reference genomes were obtained from three sources, which included the MAGs recovered from the metagenomes of this study, the published genomes downloaded from the NCBI GenBank database (37) (August 16th, 2022), and those from the genomic catalog of Earth's microbiomes (38).

## Gene annotation and metabolic pathway reconstruction

The protein-coding genes were predicted by Prodigal (v.2.6.3) (39) using "-p meta" option. Orthologous gene families were identified by OrthoFinder (v.2.2.1) (40) with the parameters "-S diamond -M msa." Gene search and annotation of MAGs analyzed in this study were performed using the Kyoto Encyclopedia of Genes and Genomes (KEGG) (41), followed by searching against Clusters of Orthologous Groups of proteins (COG) (42), the Protein families database (Pfam) (43), the Conserved Domain Database (CDD) (44), the Protein Clusters (PRK) (https://www.ncbi.nlm.nih.gov/proteinclusters/), the Institute for Genomic Research's database of protein FAMilies (TIGRFAM) (45), the carbohydrate-active enzymes database (CAZy) (46), the bacterial peptidases database (MEROPS) (47), and the NCBI non-redundant (nr) database using the BLASTP program (e-value $<1 \times 10^{-5}$, identity >30%, query coverage >50%) (48). Extracellular peptidases were predicted using SignalP (v.6.0) (49) and PSORTb (v.3.0.3) (50). The metabolic pathways were reconstructed based on the above-predicted annotations, and reference pathways were depicted in KEGG and MetaCyc (51). A detailed workflow was shown in Fig. S2. Taxonomic classification of each protein-coding gene in the *nifH*-containing contig of *Ketobacter* W.bin6.184 was identified using CAT tool (v.6.0.1) (52) against NCBI reference protein database.

## Phylogenetic analysis and HGT gene identification

The phylogenetic trees of *nxrA*, *aclA*, *aclB*, *rbcL*, *nifH*, and cytochrome *c* oxidase genes were constructed using IQ-TREE (v.1.6.3) (53) with ModelFinder (54), and ultrafast bootstrapping was used to estimate the reliability of each branch with 1,000 times resampling. The reference sequences with BLASTP identity >30% to the target genes were retrieved from the NCBI nr database (37) and the UniProt database (55). The phylogenomic tree of anammox bacteria was constructed using IQ-TREE with Model-Finder, based on the 120 conserved single-copy ubiquitous bacterial genes (33). The trees were visualized using the iTOL online tool (v.6) (56). The HGT genes were predicted by the HGTector tool (v.2.0b3) (57) and the genes encoding type A cytochrome *c* oxidase in D200.bin4.133 were analyzed by constructing a phylogenetic tree using IQ-TREE.

## RESULTS

### Microbial community structures in the MT biosphere

A detailed survey based on the 16S rRNA gene sequences retrieved from the clean reads of the metagenome revealed 348, 222, and 139 microbial classes in the deep sediments, surface sediments, and seawater of the MT, respectively (Fig. 1; Table S2). The three habitats all showed a dominance of bacteria over archaea (>84.5% bacteria in relative abundance). *Proteobacteria*, *Chloroflexota*, *Planctomycetota,* and *Thermoproteota* (represented by *Nitrososphaeria*) were common microbial groups in the MT

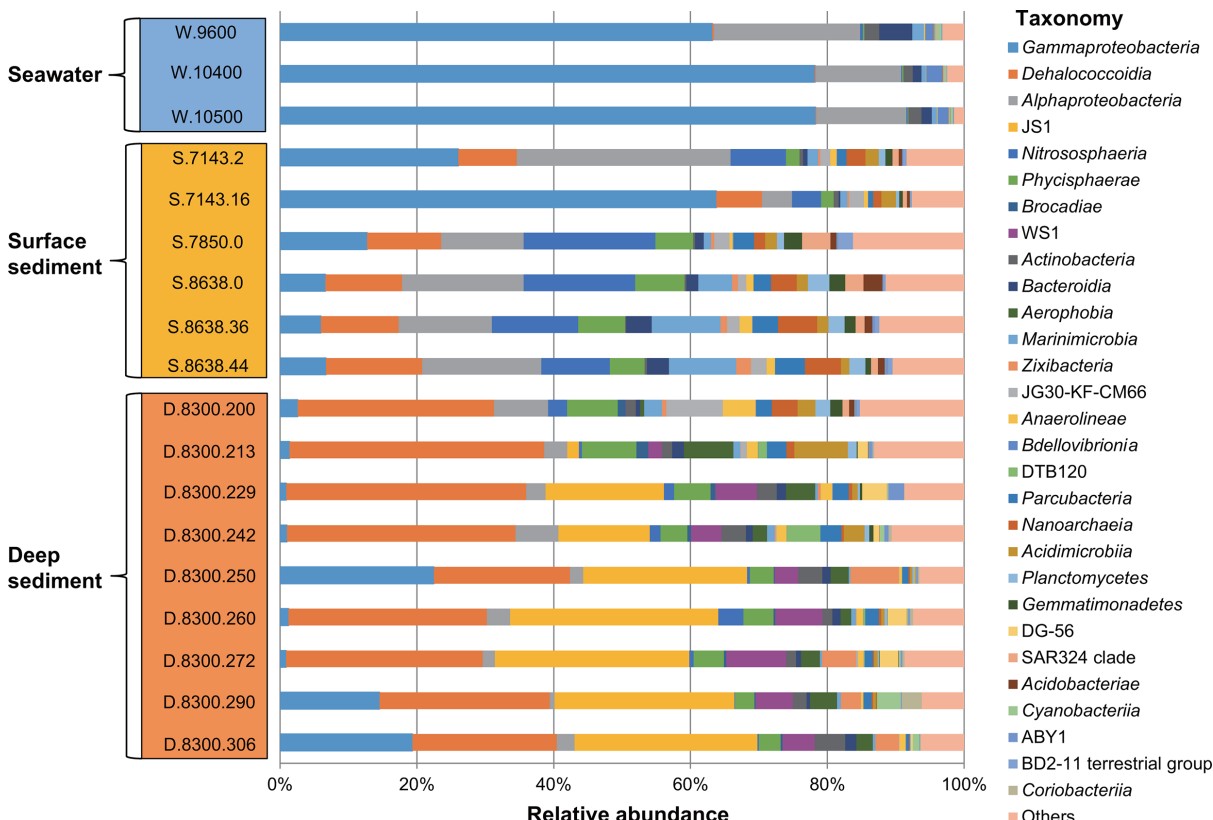

**FIG 1** Relative abundances of dominant microbial groups based on 16S rRNA gene sequences retrieved from the clean reads of each metagenome. A relative abundance of less than 2% was grouped into others. Detailed values are provided in Table S2.

sediments and seawater. However, distinct dominant microbial taxa were observed in the three habitats (Fig. 1; Table S2). For example, *Dehalococcoidia* (*Chloroflexota*) (>19%) and JS1 (*Atribacterota*) (>1.6%) were the predominant microbial groups in the deep sediments. The highest relative abundances of *Dehalococcoidia* occurred in each sample of D.8300.200 (28.6%), D.8300.213 (37.0%), D.8300.229 (34.5%), D.8300.242 (31.7%), and D.8300.272 (28.2%). The JS1 bacteria had the highest relative abundance in each sample of D.8300.250 (23.4%), D.8300.260 (29.9%), D.8300.290 (25.2%), and D.8300.306 (25.5%). Interestingly, the average relative abundance of *Brocadiae* (*Planctomycetota*) in the MT deep sediments (200–306 cmbsf) was significantly higher than that in surface sediments (0–46 cmbsf) (0.57% versus 0.03%; Mann-Whitney U test, $P < 0.01$).

*Alphaproteobacteria* was a major group in the surface sediment at S.7143.2 (31.1%), S.7143.16 (4.4%), S.7850.0 (12.0%), S.8638.0 (17.7%), S.8638.36 (13.6%), and S.8638.44 (17.4%). Furthermore, a higher relative abundance of *Nitrososphaeria* was detected in the surface sediment compared with the deep sediment or seawater. In the seawater, the major microbial groups included *Gammaproteobacteria* (60.3-73.7%) and *Alphaproteobacteria* (11.8%–20.4%).

## Diverse microbial nitrogen cycling processes

In general, various microorganisms contain key genes that participate in nitrogen cycling processes, including DNRA [membrane-bound nitrate reductase (NAR, *narG*), periplasmic nitrate reductase (NAP, *napA*), periplasmic cytochrome nitrite reductase (ccNIR, *nrfA*)], denitrification [NAR, NAP, heme-containing nitrite reductase (cd$_1$-NIR, *nirS*), Cu-containing nitrite reductase (Cu-NIR, *nirK*), cytochrome *c*-dependent nitric oxide reductase (cNOR, *norB*), quinol-dependent nitric oxide reductase (qNOR, *norZ*), nitrous oxide reductase (NOS, *nosZ*)], nitrification [ammonia monooxygenase (AMO,

*amoA*), hydroxylamine oxidoreductase (HAO, *hao*), nitrite oxidoreductase (NXR, *nxr*)], anammox [hydrazine dehydrogenase (HAO, *hao*), hydrazine synthase (HZS, *hzs*)], and nitrogen fixation [molybdenum-iron nitrogenase (MoFe, *nifH*)] (17, 22). The abundances of functional genes [expressed as reads per kilobase per million sequenced reads (RPKM)] in the denitrification pathway (*nirK*, *norBC*) were higher in the surface sediments than in the deep sediments. An abundance of genes in the DNRA or denitrification process (*napA/narG*, *nrfA*) was higher in the surface sediments than in the seawater (Fig. 2; Table S3; Mann-Whitney U test, $P < 0.05$). The MT seawater metagenomes also showed higher abundances of *nirK* and *norB* genes, as compared with those in the deep sediments (Mann-Whitney U test, $P < 0.05$). However, the abundance of *nosZ* gene in the deep sediments was eightfold higher than in seawater (Mann-Whitney U test, $P < 0.05$). Notably, unique microbial nitrogen cycling genes were detected in the three habitats,

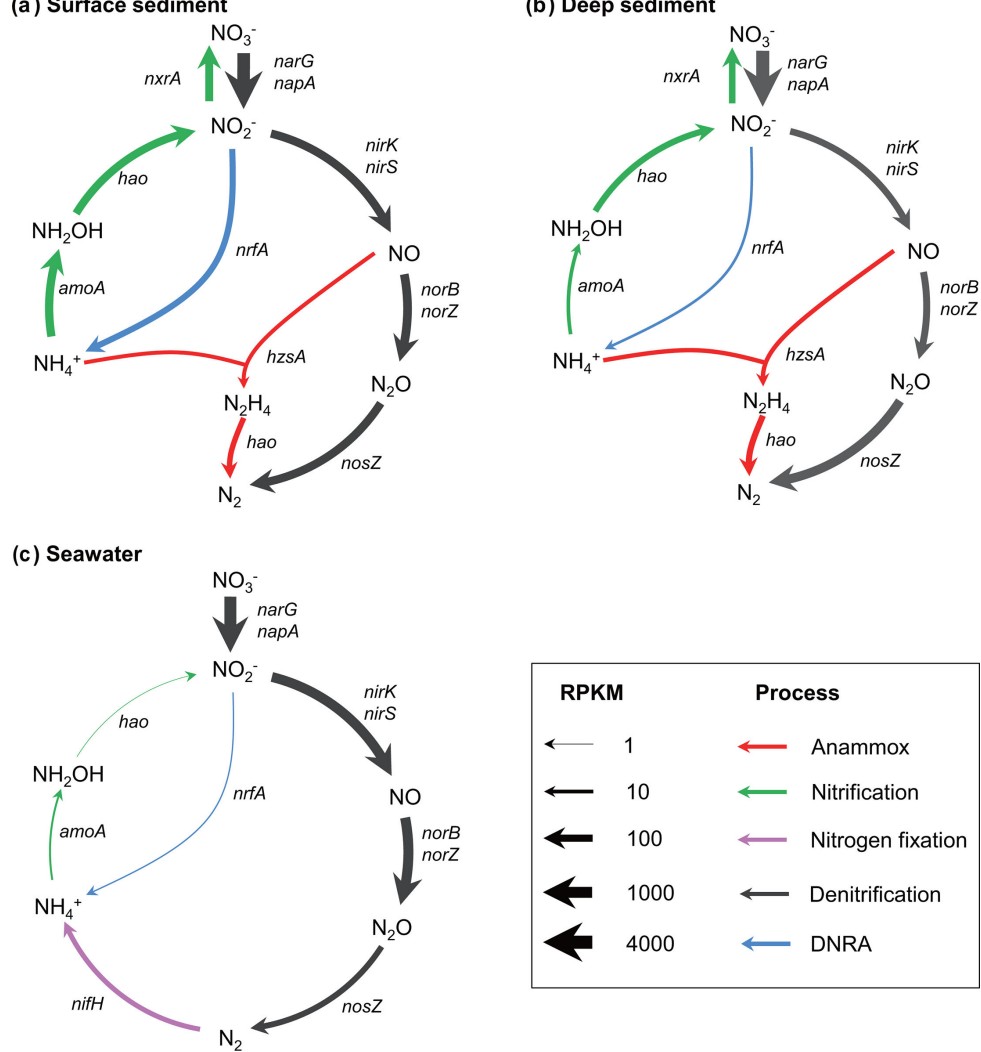

**FIG 2** Microbial nitrogen cycling networks in the (a) surface sediment, (b) deep sediment, and (c) seawater of the MT. The thickness of each line represents the average abundance (expressed as RPKM) of the key genes in the pathway. Different colors represent different microbial nitrogen processes. *narG*, membrane-bound nitrate reductase alpha subunit; *napA*, periplasmic nitrate reductase; *nrfA*, periplasmic cytochrome *c* nitrite reductase, *nirS*, heme-containing nitrite reductase; *nirK*, Cu-containing nitrite reductase; *norB*, cytochrome *c*-dependent nitric oxide reductase subunit B; *norZ*, quinol-dependent nitric oxide reductase; *nosZ*, nitrous oxide reductase; *amoA*, ammonia monooxygenase subunit A; *hao*, hydroxylamine oxidoreductase/hydrazine dehydrogenase; *nxrA*, nitrite oxidoreductase alpha subunit; *hzsA*, hydrazine synthase subunit A; *nifH*, molybdenum-iron nitrogenase. Detailed values are provided in Table S3.

including nitrite oxidation and anammox in the sediments, and nitrogen fixation in the seawater.

## Newly-identified NOB and anammox bacteria in the MT sediment

Nitrite oxidation was a significant microbial nitrogen process in the MT sediment compared with seawater because the metagenomes in the surface and deep sediments contained the hallmark gene nitrite oxidoreductase (*nxr*) (average RPKM value of 149 and 60, respectively; Fig. 2; Fig. S3). Of note, seven nitrite oxidoreductase-containing genomes were recovered from the metagenomes of the MT deep sediment. These MAGs were assigned to taxa including *Desulfobacterota*, *Latescibacterota*, *Omnitrophota*, *Planctomycetota*, and unclassified bacteria JABMQX01 according to the taxonomy of Genome Taxonomy Database (GTDB) (Table S4), which are distinct from the common NOB taxa (e.g., *Chloroflexota*, *Proteobacteria*, *Nitrospinota*, and *Nitrospirota*) as described in previous studies (58). Furthermore, these NOBs could potentially fix inorganic carbon because the complete gene sets in the rTCA or CBB cycles were identified in the NOB genomes, including the key genes encoding ATP citrate lyase (*acl*) or ribulose-bisphosphate carboxylase (*rbc*), respectively (Fig. 3; Fig. S4 through S6; Table S5).

Furthermore, anammox, determined by the presence of the hydrazine synthase gene (*hzs*), only occurred in the MT sediment (Fig. 2). Three anammox MAGs were then retrieved from the metagenomes of the MT deep sediments, among which D.213.bin9.96 and D.200.bin4.133 belonged to *Scalinduaceae*, and D.213.bin8.3 belonged to the recently named *Bathyanammoxibiaceae* (Fig. S6). To reveal the characteristics of anammox bacteria in the MT deep sediment, we performed the comparative genomic analysis of 76 anammox bacteria genomes (>50% completeness and <5% contamination), including 3 MAGs retrieved from this study (59, 60), 69 MAGs from the NCBI Assembly database (37), and 4 MAGs from the genomic catalog of Earth's microbiomes (38). Interestingly, some genomes from *Bathyanammoxibiaceae*, *Scalinduaceae,* and *Brocadiaceae* contained genes encoding cytochrome *c* oxidases (Fig. S7). A phylogenetic analysis of the cytochrome *c* oxidase large subunit genes showed that D.200.bin4.133 belonged to type A cytochrome *c* oxidase (*aa3*) and that genes in the other anammox

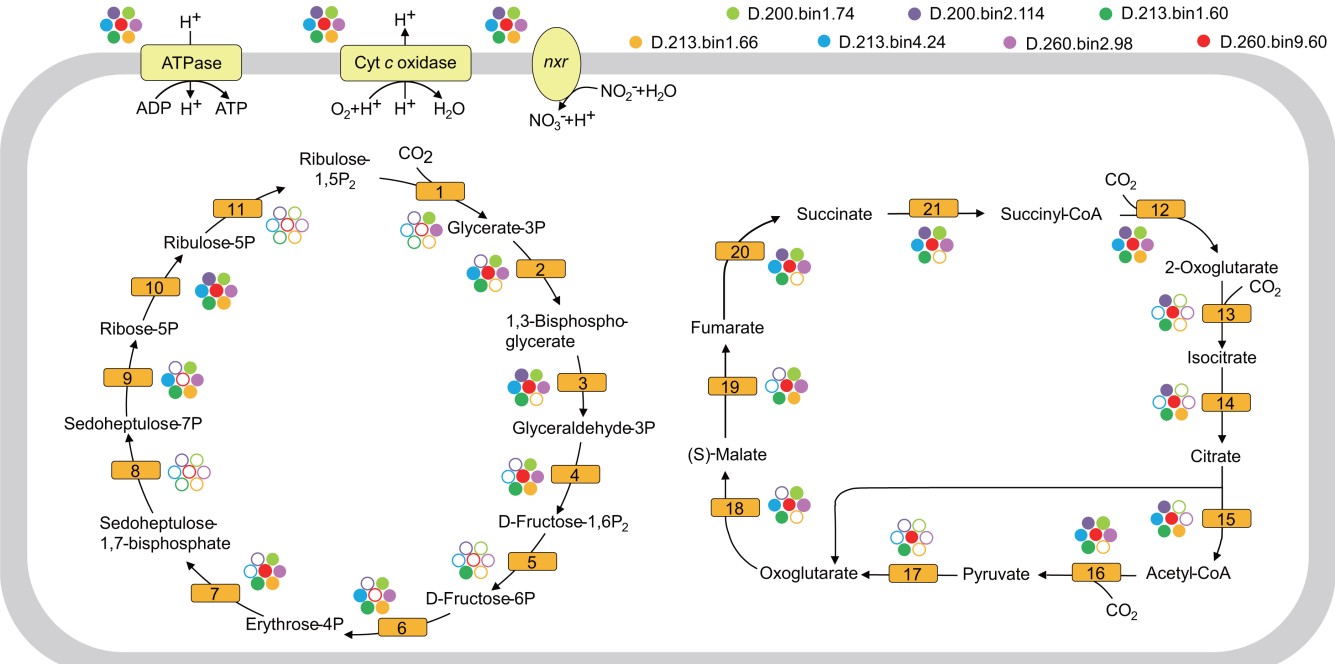

**FIG 3** Reconstructed metabolic pathways of seven NOB MAGs in the MT deep sediment. Each filled or hollow circle indicates that the gene is present or absent in the MAG, respectively. A full list of genes labeled with different letters is provided in Table S5.

bacteria were type C cytochrome *c* oxidase (*cbb3*) (Fig. S8). In addition, complete gene sets of the glycolysis pathway were detected in the three anammox MAGs from the MT deep sediments, including those encoding enzymes that oxidize multiple carbohydrates such as glucose, lactose, starch/glycogen, and oligosaccharides (Fig. 4; Table S6).

## N₂ fixers in the MT seawater

The key gene *nifH*, which encodes nitrogenase, was detected exclusively in the MT seawater, suggesting the existence of microorganisms that convert inert N₂ to bioavailable ammonia. Furthermore, the metagenomic binning of the three seawater samples recovered a *nifH*-containing MAG W.bin6.184. The MAG was assigned to the genus *Ketobacter* of the family *Alcanivoracaceae*, and its average relative abundance was 4.0% in the MT seawater (Fig. 2c; Fig. S9; Table S4). Meanwhile, all of the protein-coding genes in the *nifH*-containing contig of W.bin6.184 were assigned to *Ketobacter* (Table S7). To assess the metabolic features of W.bin6.184, we performed a comparative genomic analysis of 29 *Ketobacter* genomes (>50% completeness and <5% contamination) retrieved from this study (W.bin6.184, W.bin6.104), the NCBI Assembly database (20 MAGs) (37), and the genomic catalog of Earth's microbiomes (7 MAGs) (38). Notably, a key gene alkane 1-monooxygenase (*alkB*), which is involved in the first step of medium-chain alkane oxidation, was detected in the two *Ketobacter* MAGs in this study (Fig. 5). In addition, complete gene sets involved in β-oxidation, the tricarboxylic acid cycle, and cytochrome *c* oxidases were present in the MAGs of W.bin6.184 (completeness 98.89%, contamination 2.39%) and W.bin6.104 (completeness 80.90%, contamination 0.65%) (Fig. 5; Table S8).

## DISCUSSION

### Distinct microbial communities exist the different MT environments

Due to the funneling effect of the hadal trench, the abundant organics accumulate in the MT trench that harbors abundant microorganisms (6). *Proteobacteria* (represented by *Gammaproteobacteria* and *Alphaproteobacteria*) and *Thaumarchaeota* (represented by *Nitrososphaeria*) were the most abundant bacterial and archaeal groups in the MT

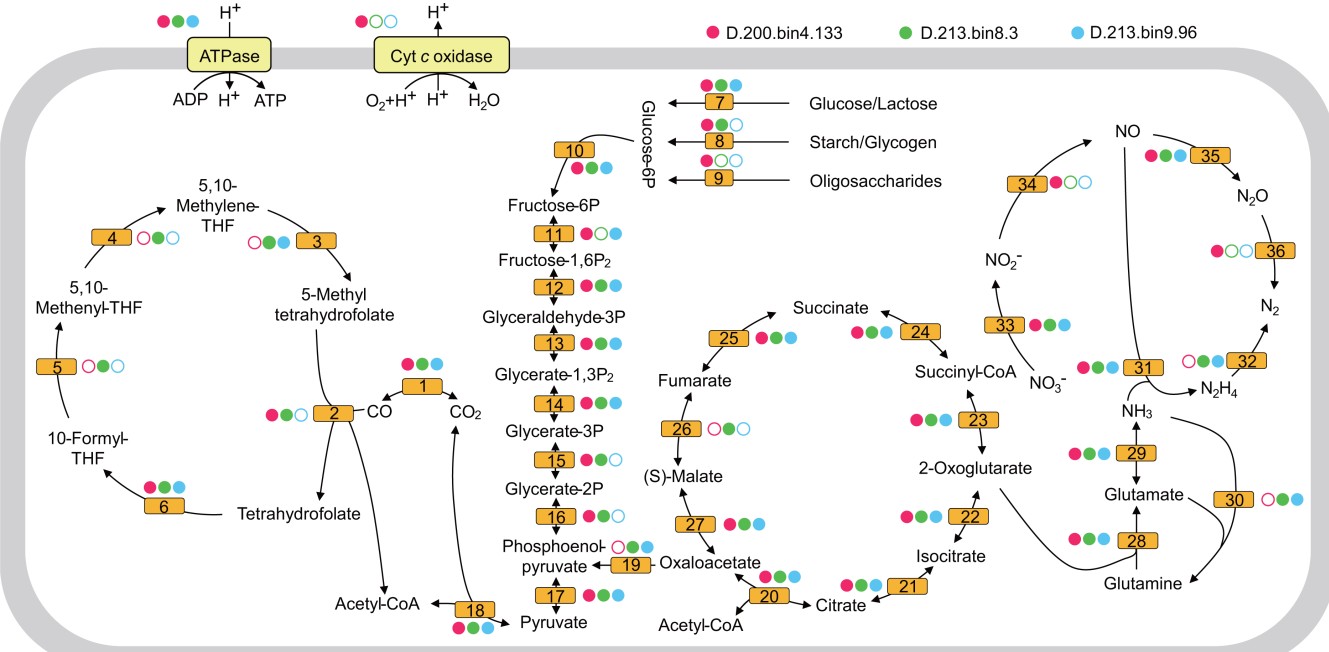

**FIG 4** Reconstructed metabolic pathways of three anammox bacteria MAGs in the MT deep sediment. Each filled or hollow circle indicates that the gene is present or absent in the MAG, respectively. A full list of genes labeled with different numbers is provided in Table S6.

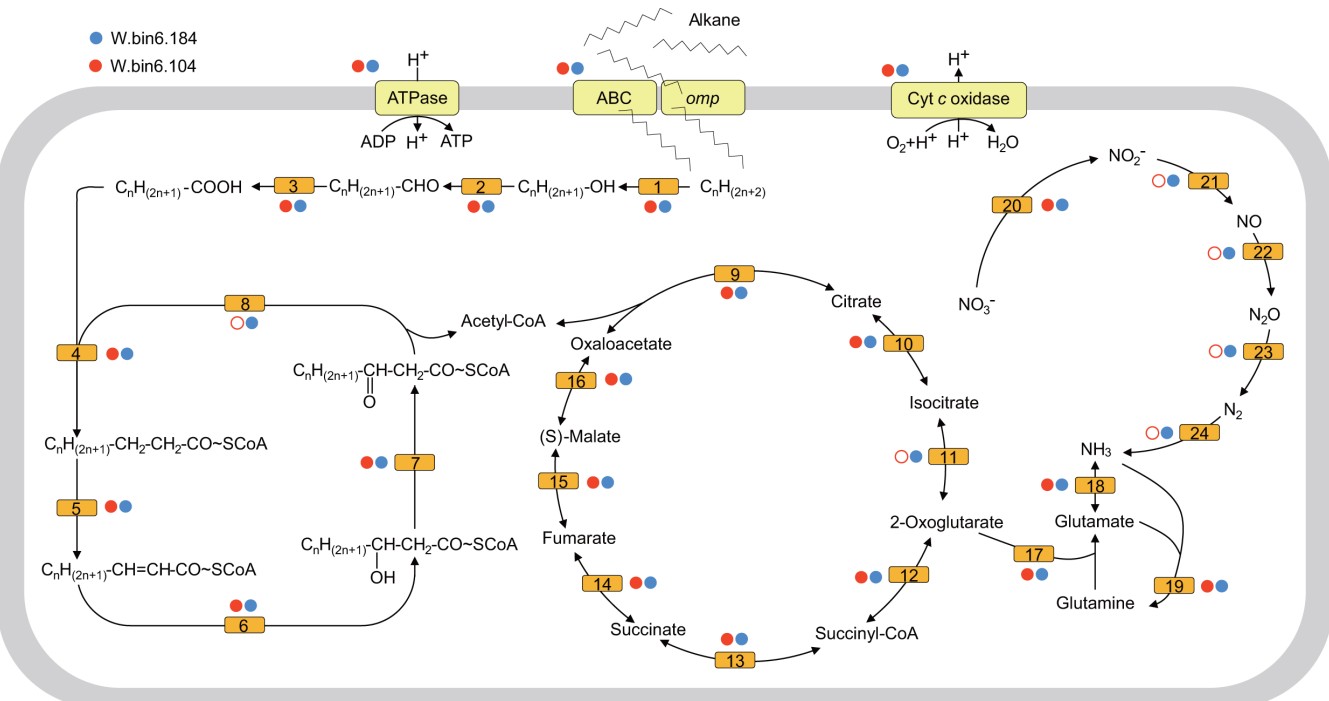

**FIG 5** Reconstructed metabolic pathways of two *Ketobacter* MAGs in the MT seawater. Each filled or hollow circle indicates that the gene is present or absent in the MAG, respectively. A full list of genes labeled with different numbers is provided in Table S8.

seawater and surface sediments, respectively (Fig. 1) (6, 25, 26). *Dehalococcoidia* and JS1 (*Atribacterota*) showed the highest relative abundances in the deep MT sediments, as compared with those in the seawater and surface sediments (Fig. 1; Mann-Whitney U test, $P < 0.05$). These results were in accordance with the deep sediments of the eastern equatorial Pacific and the North Pacific gyre, indicating that these bacterial groups might degrade halogenated organic compounds and aromatics present in the deep MT sediments (13, 61, 62).

## Five new NOB lineages fix $CO_2$ in the MT sediment

Compared with sediment from the adjacent trench bottom and abyssal plain, the sediment layer of the MT slope is generally thinner with a lower organic carbon content (4, 6). Such an environment might favor the growth of chemolithoautotrophic microorganisms, such as AOA and NOB. In general, AOA is a dominant archaeal group in many marine sedimentary environments (63). However, NOB lineages are often overlooked because of the low $NO_2^-$ concentration and its rapid turnover rate in the sediments (61, 64, 65). Previous studies reported that NOBs have a very high affinity for $NO_2^-$ and that they are the predominant fixers of dissolved inorganic carbon in the dark ocean (64, 66). In the sediment of the MT slope, the average gene abundance of *nxrA* was comparable with that of *norBZ* (Fig. 2), suggesting that nitrite oxidation might be an important process that shares similar activity with NO reduction in the predominant denitrification pathway. The retention of nitrogen *via* nitrite oxidation ($NO_2^- \rightarrow NO_3^-$) reduces the loss of nitrogen *via* denitrification ($NO_2^- \rightarrow NO$) (67). However, the source of $NO_2^-$ in the MT sediments is unknown because the concentration of $NO_2^-$ was below the detection limit (<0.1 µM) (10). A recent study showed that the microbial nitrate reduction process was stimulated in the MT sediments (68), suggesting that $NO_2^-$ is supplied to the sediment through the microbial nitrate reduction process, or from the surrounding environment (58, 69).

Seven NOB genomes were recovered from the metagenomes of the MT deep sediments (Fig. 3; Fig. S3). These MAGs were identified as new NOB lineages that are not classified into the common NOB taxa *Nitrospinota*, *Nitrospirota*, *Chloroflexota*, or *Proteobacteria*. Of note, each of the seven new NOBs could fix inorganic carbon through the rTCA or CBB cycle (Fig. 3) (70). However, the $O_2$ content in the MT sediment would be much lower than that in the surrounding seawater, indicating that these hadal NOBs had a microaerophilic lifestyle and entered the hadal sediment through physical setting or biological processes as similar to the NOB from *Nitrospira* (58, 71). As a result, the identification of these new NOBs not only extends their phylogenetic diversity but also reveals that nitrite oxidation might be an important yet overlooked microbial carbon fixation process in hadal sediment (71).

## Anammox bacteria perform aerobic respiration in the MT sediment

The identification of the *hzs* gene and the recovery of three anammox MAGs in the MT sediment confirm previous results reporting that anammox is an important nitrogen removal process in hadal sedimentary environments (6, 24, 61). Of note, the anammox bacterium (D.200.bin4.133) could perform aerobic respiration because it contained a gene encoding the type A cytochrome *c* oxidase (Fig. S7). There are three types of cytochrome *c* oxidases (type A, B, and C), of which type A cytochrome *c* oxidase mainly functions in high oxygen concentration environments compared with the other two types (72). Type A cytochrome *c* oxidase is also the terminal oxidase with the highest efficiency for energy generation (73). In some bacterial strains containing type A and type C cytochrome *c* oxidases (e.g., *Pseudomonas aeruginosa* PAO1 and *Shewanella oneidensis* MR-1), type A is highly expressed whereas type C remains unchanged under nutrient-limiting conditions (74, 75). This implies that compared with type C cytochrome *c* oxidase, type A cytochrome *c* oxidase is more adapted to oligotrophic environments such as the MT deep sediment. In addition, a phylogenetic tree containing a collection of 229 type A cytochrome *c* oxidase reference genes revealed that D.200.bin4.133 might have acquired the type A cytochrome *c* oxidase gene from *Desulfobacterota* by horizontal gene transfer (Fig. S10).

Furthermore, the metabolic pathway reconstruction of the anammox bacterium D.200.bin4.133 showed that this strain can obtain energy by utilizing multiple carbohydrates (glucose, lactose, starch/glycogen, or oligosaccharides) (Fig. 4), which would provide more ATPs than anaerobic metabolism when $O_2$ is the terminal electron acceptor (76, 77). Anammox bacteria have demonstrated their ability to live in oxygenated environments. For example, "*Candidatus* Brocadia caroliniensis" and "*Candidatus* Kuenenia stuttgartiensis" were observed to tolerate oxygen concentrations of up to 120 µM and 200 µM, respectively (78). The defense mechanisms against oxygen in these anammox bacteria may be attributed to the expression of enzymes such as bilirubin oxidase, cytochrome *c* oxidase, and bifunctional catalase-peroxidase (79). Similarly, the gene encoding type A cytochrome *c* oxidase in the genome of D.200.bin4.133 might act as an oxygen scavenger to maintain the anammox process in an anaerobic environment, which confers a selective advantage to D.200.bin4.133 that faces $O_2$ concentration fluctuations in MT (4, 80, 81). The $O_2$ content in the MT seawater below 6,000 mbsl was constantly within the range between 156 µM and 188 µM (10). The source of $O_2$ in the MT sediment may be derived from the overlying seawater that penetrates through the seawater-sediment interface (6, 82). Moreover, the abundant AOA species in the MT sediment might also produce a small amount of $O_2$ as reported recently (83, 84), and the $O_2$ could be rapidly utilized by other aerobic microorganisms such as anammox bacterium D.200.bin4.133. However, more evidence is needed to test the activity of aerobic respiration of MT anammox bacteria, such as metatranscriptomics, metaproteomics, and rate measurements.

## Alkane-degrading *Ketobacter* fix N$_2$ in the MT seawater

N$_2$ production by microbial denitrification and anammox processes forms the largest nitrogen sink in the ocean (24, 85). Recent studies revealed that most of N$_2$ is produced *via* the anammox process in the hadal sediments of the Atacama Trench (~67%) and Kermadec Trench (>90%) (6, 24). By contrast, in the MT seawater, the denitrification process is responsible for most N$_2$ production because nitrate (~36 µM) and denitrifying microorganisms are abundant (Fig. 2) (10, 25). The accumulated N$_2$ is then used by an abundant nitrogenase-containing *Ketobacter* strain W.bin6.184 (Fig. 5). However, N$_2$ fixation is an extremely high energy-consuming process (16 ATPs per mole of N$_2$ fixed) (86, 87). Various organic matter could serve as energy sources for nitrogen fixation, such as cellulose, chitin, glucan, pectin, polyphenols, starch, and alkane (88, 89). To fulfill this large energy requirement, *Ketobacter* W.bin6.184 might have the potential to perform the aerobic degradation of medium-chain alkanes to produce acetyl-CoA (>100 ATPs per mole of medium-chain alkane oxidized) (90) because of the presence of complete gene sets for alkane oxidation (*alk* and β-oxidation) (Fig. 5). This is consistent with the features of most of the genera in *Alcanivoracaceae*, a well-known aerobic hydrocarbon-degrading bacterial family (91, 92). Of note, the *alkB* gene was detected in most of the *Ketobacter* genomes (Table S8), and the utilization of n-alkane by a pure *Ketobacter* strain was demonstrated by incubation experiments (91), suggesting that *Ketobacter* W.bin6.184 could obtain energy from alkane degradation. The concentration of n-alkanes was 23.5 µg/gdw in the MT seawater as measured previously (26, 93), and the alkane-degrading bacteria including the members of *Alcanivoracaceae* were abundant in the MT seawater (Fig. 1). Possible sources of the alkanes in the MT might include a mixture of biological processes (through rotting organisms) and geological processes (through water-rock reactions) (94, 95). The slow degradation of alkanes may spread long distances with the effects of hadal seawater currents (94).

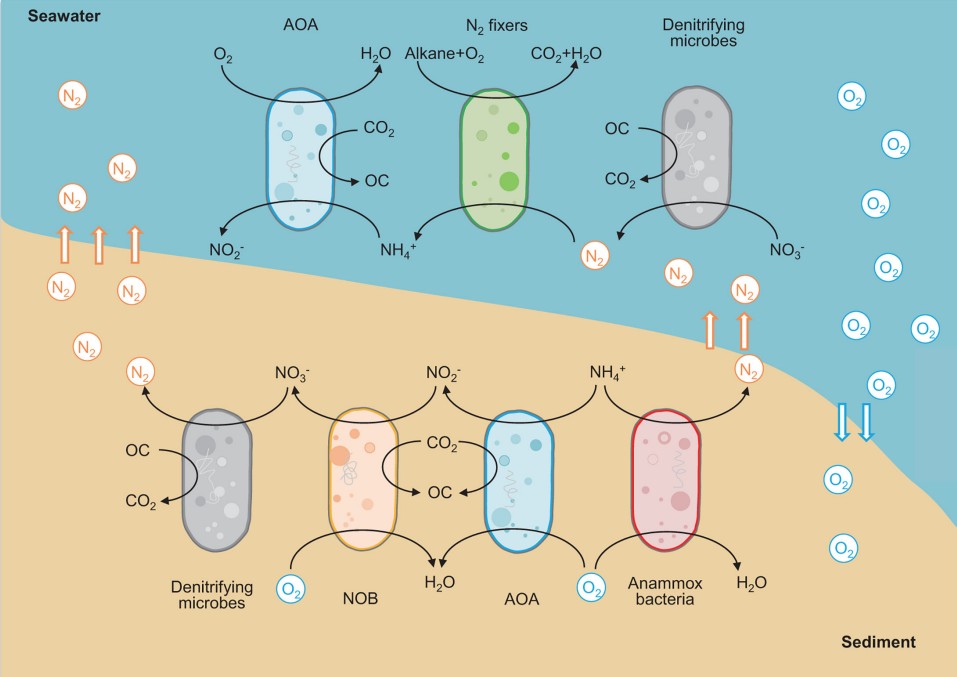

**FIG 6** Schematic representation of the microbial nitrogen cycling processes in the MT sediment and seawater. OC, organic carbon. Detailed information on the MAGs belonging to AOA, NOB, anammox bacteria, N$_2$ fixers, and denitrifying microbes is provided in Table S4.

## Conclusion

This study provides new insights into the unique features of the microbial nitrogen cycling processes in the deepest part of the ocean. The distinct dominant microbial taxa were observed in the different MT habitats. The identification of five new NOB lineages in the MT sediment has uncovered an overlooked process for inorganic carbon fixation. Meanwhile, anammox bacteria might perform aerobic respiration in response to nutrient limitations or $O_2$ fluctuations in the sediment. In the MT seawater, an abundant *Ketobacter* strain might obtain energy during alkane degradation, and then fix $N_2$ released from sedimentary denitrifiers and anammox bacteria (Fig. 6). The integration of multi-omics strategy that combines metagenomics with metatranscriptomics, metaproteomics, or metabolomics will provide more comprehensive understanding of hadal microbial communities in future studies. Meanwhile, laboratory incubation experiments and *in situ* activity tests are also needed to verify the contribution of hadal microbial communities to the global nitrogen biogeochemical cycles.

## ACKNOWLEDGMENTS

We thank Wan Liu for the genome submission to eLMSG (https://www.biosino.org/elmsg/index).

Y.H.: Conceptualization, Formal analysis, Visualization, Writing—Original Draft. X.Z.: Conceptualization, Methodology, Software, Resources, Investigation, Writing—Original Draft. Y.X.: Methodology, Writing—Review & Editing. J.T.: Methodology, Writing—Review & Editing. M.L.: Supervision, Conceptualization, Writing-Review & Editing, Funding acquisition.

## AUTHOR AFFILIATIONS

[1]Archaeal Biology Center, Shenzhen Key Laboratory of Marine Microbiome Engineering, Institute for Advanced Study, Shenzhen University, Shenzhen, China
[2]Synthetic Biology Research Center, Institute for Advanced Study, Shenzhen University, Shenzhen, China
[3]College of Life Sciences and Oceanography, Shenzhen University, Shenzhen, China
[4]Key Laboratory of Marine Chemistry Theory and Technology, Ministry of Education, Institute for Advanced Ocean Study, Ocean University of China, Qingdao, Shandong, China
[5]MOE Key Laboratory of Physical Oceanography, Frontiers Science Center for Deep Ocean Multispheres and Earth System, Ocean University of China, Qingdao, China

## AUTHOR ORCIDs

Yuhan Huang  http://orcid.org/0000-0001-8675-0758
Xinxu Zhang  http://orcid.org/0000-0002-4299-2795
Meng Li  http://orcid.org/0000-0001-8675-0758

## FUNDING

| Funder | Grant(s) | Author(s) |
| --- | --- | --- |
| MOST \| National Natural Science Foundation of China (NSFC) | 91951102 | Xinxu Zhang |
| MOST \| National Natural Science Foundation of China (NSFC) | 92251306, 32070108, 32225003 | Meng Li |
| Guangdong Major Project of Basic and Applied Basic Research | 2023B0303000017 | Meng Li |
| Southern Marine Science and Engineering Guangdong Laboratory (Zhuhai) | SML2023SP218 | Meng Li |

| Funder | Grant(s) | Author(s) |
|---|---|---|
| Shenzhen University 2035 Program for Excellent Research | 2022B002 | Meng Li |
| Synthetic Biology Research Center of Shenzhen University | | Meng Li |

## AUTHOR CONTRIBUTIONS

Meng Li, Funding acquisition, Supervision, Writing – review and editing.

## DATA AVAILABILITY

The genome sequences from the current study have been deposited in eLMSG (an eLibrary of Microbial Systematics and Genomics, https://www.biosino.org/elmsg/) under accession numbers (MAGs) LMSG_G000011485.1-LMSG_G000011652.1. Raw reads of the metagenomes for MT deep sediments have been deposited in NODE (The National Omics Data Encyclopedia, https://www.biosino.org/node/) under the accession number OES290491-OES290499.

## ADDITIONAL FILES

The following material is available online.

### Supplemental Material

**Supplemental Figures (mSystems00243-24-s0001.pdf).** Figures S1 to S10.
**Supplemental Tables (mSystems00243-24-s0002.xlsx).** Tables S1 to S8.

### Open Peer Review

**PEER REVIEW HISTORY (review-history.pdf).** An accounting of the reviewer comments and feedback.

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
