## [Reviewer comments · mSystems]

Distinct microbial nitrogen cycling processes in the deepest part of the ocean

Yuhan Huang, XinXu Zhang, Yu Xin, Jiwei Tian, and Meng Li

Corresponding Author(s): Meng Li, Shenzhen University

Review Timeline:

Submission Date:	February 18, 2024
Editorial Decision:	April 17, 2024
Revision Received:	May 22, 2024
Accepted:	June 1, 2024

Editor: Haiyan Chu

Reviewer(s): Disclosure of reviewer identity is with reference to reviewer comments included in decision letter(s). The following individuals involved in review of your submission have agreed to reveal their identity: Hongchen Jiang (Reviewer #1)

Transaction Report:

DOI: <https://doi.org/10.1128/msystems.00243-24>

Re: mSystems00243-24 (Distinct microbial nitrogen cycling processes in the deepest part of the ocean)

Dear Prof. Meng Li:

Revision Guidelines

Sincerely,
Haiyan Chu
Editor
mSystems

Reviewer #1 (Comments for the Author):

This study conducted a comparative genomics analysis on microbial nitrogen cycling in the Mariana Trench (MT) and examined the nitrogen cycling of microorganisms in three different MT habitats. This research discovered five new lineages of nitrite-oxidizing bacteria (NOB) and revealed how they adapt to the oligotrophic MT slope sediments. In addition, the study also found that the abundant alkane-oxidizing *Ketobacter* species in seawater can utilize inert N₂ released by other denitrifying and/or anaerobic ammonia-oxidizing bacteria. These findings are fantastically interesting and expand our understanding of microbial life

in the deep sea. Therefore, the content of this study deserves publication. Before that, the manuscript needs some revisions.

Major Review Comments:

1. The manuscript needs to provide a clearer description of the research methods and experimental procedures to ensure that readers can accurately understand the design and results of this study.
2. The results and discussion sections require more detailed validation. When cultivation limitations exist, referring culture strains with similar functions for validation experiments may support the authors' conclusions and findings.
3. The discussion section could further explore the significance (implication) of these new discoveries for the nitrogen cycling and deep-sea microbial ecosystems, as well as the directions for future research and possible applications.

Minor Comments:

- 1). In lines 172-174, it is recommended to use higher completeness Metagenome-Assembled Genomes (MAGs) for further analysis.
- 2). In lines 188-190, please clearly explain why the Mann-Whitney U test was used and its statistical significance.
- 3). In lines 280-293, carefully consider the descriptions. For example, lines 280-283, the process may be due to changes in oxygen concentration, and it cannot be ruled out that microbes from the surface enter the deep sea through physical settling or biological processes, gradually being selected to cause the observed situation.
- 4). In lines 299-304, it is recommended to use higher completeness genomes or assemble into scaffolds rather than contigs.
- 5). In lines 324-329, exercise caution when dealing with functional genes found only in a single genome, as this may be due to contamination. Alternatively, demonstrate how contamination in this genome was excluded.

Reviewer #2 (Comments for the Author):

The study presents novel findings, including the identification of new chemolithoautotrophic NOB lineages and the exploration of aerobic respiration in anammox bacteria, shedding light on previously overlooked aspects of deep-sea microbiology. Additionally, the investigation of alkane-degrading bacteria and their potential role in nitrogen fixation provides intriguing insights into the complex interactions within deep-sea ecosystems.

here are some specific revision suggestions based on the content of the text:

1. Clarify the Source of Nitrite in MT Sediments:

In the section discussing NOB lineages in MT sediment, it would be beneficial to provide further clarification on the potential sources of nitrite in the sediment. Since the concentration of nitrite is below the detection limit, it's essential to discuss potential mechanisms by which nitrite may be supplied to the sediment, such as through microbial processes or from the surrounding environment.

2. Provide Additional Experimental Evidence for Anammox Bacteria's Aerobic Respiration:

In the part discussing aerobic respiration in anammox bacteria, please provide additional experimental evidence to support this hypothesis.

3. Expand on the Relationship Between Alkane Degradation and N₂ Fixation:

In the section discussing alkane-degrading *Ketobacter* and N₂ fixation in MT seawater, consider expanding on the relationship between alkane degradation and nitrogen fixation. Providing more context on how alkane degradation contributes to the energy requirements for nitrogen fixation, and discussing potential mechanisms by which this process occurs, would enhance the understanding of this aspect of the study.

4. Discuss Potential Limitations of the Study:

It would be beneficial to include a section discussing potential limitations of the study. This could involve addressing any methodological constraints, uncertainties in data interpretation, or other factors that may have influenced the results or conclusions of the study. Acknowledging these limitations would provide a more balanced assessment of the research findings.

Response letter

(Questions in italic and responses in bold)

Reviewer #1:

1. The manuscript needs to provide a clearer description of the research methods and experimental procedures to ensure that readers can accurately understand the design and results of this study.

Response: We thank the reviewer for the comment. To enhance clarity and comprehensiveness, we have provided a detailed workflow of the metagenomic analysis in Figure S2 and added more detailed information about the research methods and experimental procedures. First, we have reviewed the standard “>50% completeness and <10% contamination” of the 498 Metagenome-Assembled Genomes with many previous studies [1-3], and the standard is appropriate to represent the majority of microbial genomes from a metagenome. This information is now added in Lines 174-176. Second, we have explained why the Mann-Whitney U test is used for statistical analysis in Lines 189-192. The decision is predicated upon its suitability for analyzing non-normally distributed data, thereby facilitating robust inferential conclusions within the context of our investigation. Third, we have added the method in Lines 217-219 for testing correct binning through taxonomic classification of each protein-coding gene in the contig using the CAT tool (v.6.0.1) [4].

2. The results and discussion sections require more detailed validation. When cultivation limitations exist, referring culture strains with similar functions for validation experiments may support the authors' conclusions and findings.

Response: We have added the following information to support our findings by other validation experiments. First, in section 4.3, a previous study showed that the bilirubin oxidase in an anammox bacterium “*Candidatus Loosdrechtia aerotolerans*” was involved in oxygen resistance by *in vitro* experiments [5]. The

cytochrome *c* oxidase and bifunctional catalase peroxidase were also detected in *Ca. Kuenenis* and *Ca. Scalindua*, which were possibly related to oxygen defense [6]. We have added the information in Lines 408-418. Second, in section 4.4, comparative genomic analysis revealed the prevalence of *alkB* and *nifH* genes within many genomes of the genus *Ketobacter*. The utilization of alkanes was demonstrated by culture experiments in the *Ketobacter* strains. We have provided the information in Lines 437-439 and Lines 445-448.

3. The discussion section could further explore the significance (implication) of these new discoveries for the nitrogen cycling and deep-sea microbial ecosystems, as well as the directions for future research and possible applications.

Response: According to the reviewer's suggestion, we have added such discussion in Lines 465-470. The integration of multi-omics strategy that combines metagenomics with metatranscriptomics, metaproteomics or metabolomics will provide more comprehensive understanding of the adaption mechanisms of nitrogen-cycling microorganisms in future studies. Meanwhile, laboratory incubation experiments and *in situ* activity tests are also needed to verify the contribution of hadal microbial communities to the global nitrogen biogeochemical cycles.

Minor Comments:

1). In lines 172-174, it is recommended to use higher completeness Metagenome-Assembled Genomes (MAGs) for further analysis.

Response: We have reviewed the standard “>50% completeness and <10% contamination” of the 498 Metagenome-Assembled Genomes with many previous studies [1-3] in Lines 174-176, and the standard is appropriate to represent the majority of genomes from a metagenome. Meanwhile, the completeness of most of the MAGs for the anammox bacteria, NOB and nitrogen fixers were more than 70%, as shown in Table S4.

2). *In lines 188-190, please clearly explain why the Mann-Whitney U test was used and its statistical significance.*

Response: The reason of using Mann-Whitney U test is as follow: First, the data distribution of the samples did not adhere to the normal distribution, rendering parametric tests unsuitable. Secondly, data were not paired between these samples. Thirdly, the sample sizes of each group were relatively small, further supporting the use of non-parametric Mann-Whitney U test. Meanwhile, this method is also commonly used by other studies [7, 8]. We have added the information and references in Lines 189-192.

3). *In lines 280-293, carefully consider the descriptions. For example, lines 280-283, the process may be due to changes in oxygen concentration, and it cannot be ruled out that microbes from the surface enter the deep sea through physical settling or biological processes, gradually being selected to cause the observed situation.*

Response: We have carefully considered the descriptions throughout the whole manuscript, and it has been modified as the reviewer suggested, in Lines 376-380.

4). *In lines 299-304, it is recommended to use higher completeness genomes or assemble into scaffolds rather than contigs.*

Response: We thank reviewer's comment. The completeness of the three anammox bacterial MAGs from this study exceeds 65%, as shown in Table S4. Specifically, the completeness of D.213.bin8.3 is 95.45%. The completeness of D.200.bin4.133, which harbors type A cytochrome *c* oxidase, is 77.27%.

5). *In lines 324-329, exercise caution when dealing with functional genes found only in a single genome, as this may be due to contamination. Alternatively, demonstrate how contamination in this genome was excluded.*

Response: Taxonomic classification of each protein-coding gene in the *nifH*-containing contig of *Ketobacter* W.bin6.184 has been identified using CAT tool (v.6.0.1) [4] against NCBI reference protein database (Lines 216-219). The

results show that all of the genes in the *nifH*-containing contig of *Ketobacter* W.bin6.184 are assigned to *Ketobacter*. This information is now added in Table S7 and Lines 325-327. Meanwhile, comparative genomic analysis of the 29 *Ketobacter* genomes revealed the widespread presence of both *nifH* and *alkB* genes within the genus *Ketobacter* (Lines 445-448). As a result, we conclude that *Ketobacter* W.bin6.184 (completeness 98.89%, contamination 2.39%) harbors *nifH* and *alkB* genes.

Reviewer #2 (Comments for the Author):

1. *Clarify the Source of Nitrite in MT Sediments: In the section discussing NOB lineages in MT sediment, it would be beneficial to provide further clarification on the potential sources of nitrite in the sediment. Since the concentration of nitrite is below the detection limit, it's essential to discuss potential mechanisms by which nitrite may be supplied to the sediment, such as through microbial processes or from the surrounding environment.*

Response: We thank the reviewer for the comment. A recent study showed that microbial nitrate reduction process was stimulated in the MT sediments [9], suggesting that NO_2^- is supplied to the sediment through the microbial nitrate reduction process, or from the surrounding environment [10, 11]. We have added this information in Lines 368-371.

2. *Provide Additional Experimental Evidence for Anammox Bacteria's Aerobic Respiration: In the part discussing aerobic respiration in anammox bacteria, please provide additional experimental evidence to support this hypothesis.*

Response: We thank the reviewer for the comment. We have added some evidence about the anammox bacteria's aerobic respiration in Lines 408-418. First, previous studies have shown that anammox bacteria can grow with oxygen. For example, “*Candidatus Brocadia caroliniensis*” and “*Candidatus Kuenenia stuttgartiensis*” were observed to tolerate oxygen concentrations of up to 120 μM

and 200 μM , respectively [6]. Then, some experimental evidence has shown that the defense mechanisms against oxygen in these anammox bacteria are attributed to the expression of enzymes such as bilirubin oxidase, cytochrome *c* oxidase and bifunctional catalase-peroxidase [5]. Above references support our hypothesis that the type A cytochrome *c* oxidase in anammox bacterium (D.200.bin4.133) might act as an oxygen scavenger to maintain the anammox process in an anaerobic environment, which confers a selective advantage to D.200.bin4.133 that faces O_2 concentration fluctuations in MT.

*3. Expand on the Relationship Between Alkane Degradation and N_2 Fixation: In the section discussing alkane-degrading *Ketobacter* and N_2 fixation in MT seawater, consider expanding on the relationship between alkane degradation and nitrogen fixation. Providing more context on how alkane degradation contributes to the energy requirements for nitrogen fixation, and discussing potential mechanisms by which this process occurs, would enhance the understanding of this aspect of the study.*

Response: To expand on the relationship between alkane degradation and N_2 fixation as the reviewer suggested, we have added more discussions in Lines 437-439 and Lines 445-448. Various organic matters could serve as energy sources for nitrogen fixation, such as cellulose, chitin, glucan, pectin, polyphenols, starch, and alkanes [12, 13]. Of note, the *alkB* gene was detected in most of the *Ketobacter* genomes (Table S8), and the utilization of n-alkane by a pure *Ketobacter* strain was demonstrated by incubation experiments [14], suggesting that *Ketobacter* W.bin6.184 could obtain energy from alkane degradation.

4. Discuss Potential Limitations of the Study: It would be beneficial to include a section discussing potential limitations of the study. This could involve addressing any methodological constraints, uncertainties in data interpretation, or other factors that may have influenced the results or conclusions of the study. Acknowledging these limitations would provide a more balanced assessment of the research findings.

Response: We thank the reviewer for the comments. The integration of

multi-omics that combines metagenomics with metatranscriptomics, metaproteomics or metabolomics will provide more comprehensive understanding of the adaption strategies of these nitrogen-cycling microorganisms in future studies. Meanwhile, laboratory incubation experiments and *in situ* activity tests are also needed to verify the contribution of hadal microbial communities to the global nitrogen biogeochemical cycles. This information is added in Lines 465-470.

1. Zhou YL, Mara P, Cui GJ, Edgcomb VP, Wang Y. 2022. Microbiomes in the Challenger Deep slope and bottom-axis sediments. *Nat Commun* 13:1515. <https://doi.org/10.1038/s41467-022-29144-4>.
2. Liu RL, Wei X, Song WZ, Wang L, Cao JW, Wu JW, Thomas T, Jin T, Wang ZX, Wei WX, Wei YL, Zhai HF, Yao C, Shen ZY, Du JT, Fang JS. 2022. Novel *Chloroflexi* genomes from the deepest ocean reveal metabolic strategies for the adaptation to deep-sea habitats. *Microbiome* 10:75. <https://doi.org/10.1186/s40168-022-01263-6>.
3. Chen P, Zhou H, Huang YY, Xie Z, Zhang MJ, Wei YL, Li J, Ma YW, Luo M, Ding WM, Cao JW, Jiang T, Nan P, Fang JS, Li X. 2021. Revealing the full biosphere structure and versatile metabolic functions in the deepest ocean sediment of the Challenger Deep. *Genome Biol* 22:207. <https://doi.org/10.1186/s13059-021-02408-w>.
4. von Meijenfeldt FAB, Arkhipova K, Cambuy DD, Coutinho FH, Dutilh BE. 2019. Robust taxonomic classification of uncharted microbial sequences and bins with CAT and BAT. *Genome Biol* 20:217. <https://doi.org/10.1186/s13059-019-1817-x>.
5. Yang Y, Lu Z, Azari M, Kartal B, Du H, Cai M, Herbold CW, Ding X, Denecke M, Li X, Li M, Gu JD. 2022. Discovery of a new genus of anaerobic ammonium oxidizing bacteria with a mechanism for oxygen tolerance. *Water Res* 226:119165. <https://doi.org/10.1016/j.watres.2022.119165>.
6. Oshiki M, Satoh H, Okabe S. 2016. Ecology and physiology of anaerobic

- ammonium oxidizing bacteria. *Environ Microbiol* 18:2784-2796.
<https://doi.org/10.1111/1462-2920.13134>.
7. Zhang M, Zhang XX, Tran NT, Sun ZQ, Zhang XS, Ye HH, Zhang YL, Ma HY, Aweya JJ, Li SK. 2021. Molting Alters the Microbiome, Immune Response, and Digestive Enzyme Activity in Mud Crab (*Scylla paramamosain*). *mSystems* 6:e0091721. <https://doi.org/10.1128/mSystems>.
 8. Hiraoka S, Hirai M, Matsui Y, Makabe A, Minegishi H, Tsuda M, Juliarni, Rastelli E, Danovaro R, Corinaldesi C, Kitahashi T, Tasumi E, Nishizawa M, Takai K, Nomaki H, Nunoura T. 2020. Microbial community and geochemical analyses of trans-trench sediments for understanding the roles of hadal environments. *ISEM J* 14:740-756.
<https://doi.org/10.1038/s41396-019-0564-z>.
 9. Yang N, Lv YX, Ji MK, Wu SG, Zhang Y. 2024. High hydrostatic pressure stimulates microbial nitrate reduction in hadal trench sediments under oxic conditions. *Nat Commun* 15:2473.
<https://doi.org/10.1038/s41467-024-46897-2>.
 10. Daims H, Lücker S, Wagner M. 2016. A new perspective on microbes formerly known as nitrite-oxidizing bacteria. *Trends Microbiol* 24:699-712.
<https://doi.org/10.1016/j.tim.2016.05.004>.
 11. Palatinszky M, Herbold C, Jehmlich N, Pogoda M, Han P, von Bergen M, Lagkouvardos I, Karst SM, Galushko A, Koch H, Berry D, Daims H, Wagner M. 2015. Cyanate as an energy source for nitrifiers. *Nature* 524:105-108.
<https://doi.org/10.1038/nature14856>.
 12. Chen YX, Nishihara A, Haruta S. 2021. Nitrogen-fixing ability and nitrogen fixation-related genes of thermophilic fermentative bacteria in the genus *Caldicellulosiruptor*. *Microbes Environ* 36:ME21018.
<https://doi.org/10.1264/jsme2.ME21018>.
 13. Dong XY, Zhang CW, Peng YY, Zhang HX, Shi LD, Wei GS, Hubert CRJ, Wang Y, Greening C. 2022. Phylogenetically and catabolically diverse diazotrophs reside in deep-sea cold seep sediments. *Nat Commun* 13:4885.

<https://doi.org/10.1038/s41467-022-32503-w>.

14. Kim SH, Kim JG, Jung MY, Kim SJ, Gwak JH, Yu WJ, Roh SW, Kim YH, Rhee SK. 2018. *Ketobacter alkanivorans* gen. nov., sp nov., an n-alkane-degrading bacterium isolated from seawater. *Int J Syst Evol Microbiol* 68:2258-2264. <https://doi.org/10.1099/ijsem.0.002823>.

Re: mSystems00243-24R1 (Distinct microbial nitrogen cycling processes in the deepest part of the ocean)

Dear Prof. Meng Li:

Your manuscript has been accepted, and I am forwarding it to the ASM production staff for publication. Your paper will first be checked to make sure all elements meet the technical requirements. ASM staff will contact you if anything needs to be revised before copyediting and production can begin. Otherwise, you will be notified when your proofs are ready to be viewed.

Sincerely,

Haiyan Chu
Editor
mSystems

Reviewer #1 (Comments for the Author):

I have some editing suggestions, which are shown in the attached file. Please check it out for more details.

**Distinct microbial nitrogen cycling processes in the deepest part of**
**the ocean**

Yuhan Huang,^{a,b} Xinxu Zhang,^{a,b} Yu Xin,^c Jiwei Tian,^d Meng Li^{a,b,#}

5 ^aArchaeal Biology Center, Shenzhen Key Laboratory of Marine Microbiome
Engineering, Institute for Advanced Study, Shenzhen University, Shenzhen 518060,
PR China

8 ^bSynthetic Biology Research Center, Institute for Advanced Study, Shenzhen
University, Shenzhen 518060, PR China

10 ^cKey Laboratory of Marine Chemistry Theory and Technology, Ministry of Education,
Institute for Advanced Ocean Study, Ocean University of China, Qingdao, Shandong,
China

13 ^dMOE Key Laboratory of Physical Oceanography, Frontiers Science Center for Deep
Ocean Multispheres and Earth System, Ocean University of China, Qingdao, China

Running Head: Microbial nitrogen cycling in the Mariana Trench

#Address correspondence to Meng Li, limeng848@szu.edu.cn (Meng Li)

Yuhan Huang and Xinxu Zhang contributed equally to this work. Author order was
determined by drawing straws.

**Abstract**

The Mariana Trench (MT) is the deepest part of the ocean on Earth. Previous
studies have described the microbial community structures and functional potential in
the seawater and surface sediment of MT, but the metabolic features and adaptation
strategies of the microorganisms involved in nitrogen cycling processes are poorly
understood. In this study, comparative metagenomic approaches were used to study
microbial nitrogen cycling in three MT habitats, including hadal seawater
(9,600-10,500 m below sea level (mbsl)), surface sediments (0-46 cm below seafloor
(cmbsf) at a water depth between 7,143 and 8,638 mbsl), and deep sediments
(200-306 cmbsf at a water depth of 8,300 mbsl). We identified five new lineages of

nitrite-oxidizing bacteria (NOB) that had adapted to the oligotrophic MT slope
sediment, via their ability to fix CO₂ through the reductive tricarboxylic acid (rTCA)
or Calvin-Benson-Bassham (CBB) cycle; An anammox bacterium could perform
aerobic respiration and utilize sedimentary carbohydrates for energy generation
because it contains genes encoding type A cytochrome *c* oxidase and the full
glycolysis pathway. In seawater, abundant alkane-oxidizing *Ketobacter* species can
fix inert N₂ released by other denitrifying and/or anammox bacteria. This study
expands our understanding of microbial life in the largely unexplored deepest part of
the ocean.

**Importance**

The metabolic features and adaptation strategies of nitrogen cycling
microorganisms in the deepest part of the ocean are largely unknown. This study
revealed that anammox bacteria might perform aerobic respiration in response to
nutrient deficiencies or O₂ fluctuations in the Mariana Trench sediments. Meanwhile,
an abundant alkane-oxidizing *Ketobacter* species could fix N₂ in hadal seawater. This
study provides new insights into the role of hadal microorganisms in global
biogeochemical nitrogen cycles and significantly expands our understanding of
microbial life in the largely unexplored deepest part of the ocean.

**Keywords:** anammox, deep biosphere, hadal trench, metagenomics, nitrite oxidation

**1. Introduction**

The hadal biosphere refers to a trench area with water depths greater than 6,000
56 m, accounting for 45% of the vertical depth of the ocean. This habitat is characterized
by high hydrostatic pressure (>60 MPa), low temperature (<4 °C), and relatively
isolated environments [1-3]. In general, the topography of a trench is shaped like a
funnel consisting of slope and axis area. Importantly, the high carbon turnover rates
were measured in the sediments, which contribute to the accumulation of organic
carbon and nitrogen compounds (e.g. nitrate and amino acids) [4-6]. However,

compared to the trench axis, the trench slope is steeper with a more complex
topography (e.g. horst, graben, seamount, and ridges), and it covers a much wider area
[7, 8]. Sinking organic material that is unevenly distributed along the slope can also
slide from the top to the trench bottom via ocean crust movements (such as
earthquakes) and seawater circulation. Therefore, the sediment layer of the slope is
generally thinner than that at the trench bottom and thus has a lower content of
organic matter and microorganisms [4, 6, 9].

Previous studies have shown that diverse bacterial and archaeal communities
have been detected in the MT. For example, *Proteobacteria*, *Bacteroidota*,
*Marinisomatota* and *Thaumarchaeota* were dominant groups in the hadal seawater
[10]. The common microorganisms in surface sediment were *Proteobacteria*,
*Chloroflexota*, *Planctomycetes*, *Thaumarchaeota* and *Nanoarchaeota* [9, 11].
Recently, the metabolic features and adaptation strategies of the hadal microbiome
have been uncovered using several techniques, such as genomics (metagenomics and
metatranscriptomics), *in situ* measurements, and laboratory incubations. For example,
*Thaumarchaeota* in MT seawater possessed two distinct sets of
adenosine-triphosphate (ATP) synthases (binding Na⁺ or H⁺) in response to the
extreme hadal environment [12]. *Chloroflexi* in the sediments of MT can utilize a
wide range of organic matter, and these bacteria responded to poor nutrition and a
periodically fluctuating environment with a "feast-or-famine" metabolic strategy [13].
*Labrenzia aggregate* strains isolated from the MT (0-9,600 m) had an increased
number of plasmidic genes along with the isolation depth and obtained the additional
genetic potential mainly through plasmid exchange to withstand the extremely high
hydrostatic pressure [14]. Dermacozines, a new secondary metabolite produced by the
bacterial strain *Dermacoccus abyssi* MT1.1^T (isolated from MT sediment at 10,898
88 m), has multiple activities such as free radical scavenging and cytotoxicity [15]. Some
89 *Alphaproteobacteria* isolates (e.g. *Sagittula stellata*, *Labrenzia aggregate*, *Pelagibaca*
*bermudensis*) produce dimethylsulfoniopropionate, which protects bacteria from high
hydrostatic pressure in the aphotic deep sea [16]. Taken together, the adaptation of

microbial communities to the hadal environment may contribute to their metabolic
activities and important roles in the global biogeochemical cycles of carbon and
nitrogen.

Nitrogen is a fundamental nutrient element in the environment. The microbial
nitrogen cycle is a basic metabolic process for the synthesis of biological molecules
(e.g. amino acids, proteins and nucleic acids) during microbial growth and
reproduction. However, bioavailable nitrogen (e.g. nitrate and ammonia) is scarce on
Earth, and microbially mediated nitrogen cycling processes contribute a significant
portion of bioavailable nitrogen in many environments [17, 18]. In general, the
nitrogen compounds in hadal environments consist of NO_3^- , NO_2^- , NH_4^+ and proteins.
NH_4^+ can be released through microbial processes such as deamination and protein
degradation catalyzed by ammonia lyases, deaminases and proteases [19]. Constant
concentrations of NO_3^- (34–36 μM) and NH_4^+ (<1 μM) were observed throughout the
hadal water of MT [10, 20]. However, the concentrations of these nitrogen
compounds varied with depth in the sediment of MT. Relatively high concentrations
of NO_3^- (20–36 μM) and low concentrations of NO_2^- (~0.1 μM) and NH_4^+ (0–5 μM)
were detected in the surface sediment (<50 cmbsf) [6], but these nitrogen compounds
showed the opposite trends in the deep sediment [21]. Few studies have reported
microbial nitrogen metabolism in MT surface sediment and seawater. For example,
multiple microbial groups (e.g. *Proteobacteria*, *Planctomycetota* and *Bacteroidota*)
are involved in NO_3^- reduction (dissimilatory nitrate reduction to ammonium (DNRA)
and denitrification) [6, 22, 23]. In the surface sediment of the Atacama Trench (5–10
115 cm at 8,085 m) and the Kermadec Trench (15–20 cm at 10,010 m), about 67% and 91%
of the N_2 was derived from the anammox process, respectively [24]. Furthermore,
some anammox bacterial MAGs (metagenome-assembled genomes) were retrieved
from the MT surface sediments (18–21 cm at 10,840 m) [6]. The nitrogen fixation
gene *nifH* has been detected in MT seawater (>9,600 m) [25]. For the nitrification
process, ammonia-oxidizing archaea (AOA), which are dominated by
*Thaumarchaeota* in hadal trenches perform autotrophic carbon fixation through

ammonium oxidation [12]. However, the metabolic features and adaption strategies of
these anammox bacteria, other types of nitrifying bacteria (e.g. NOB), and N₂ fixers
have not yet been discovered in the hadal biosphere.

To address these knowledge gaps, we used comparative metagenomic approaches

[revised manuscript text omitted]

the dominant denitrification pathway. Nitrogen retention via nitrite oxidation
($\text{NO}_2^- \rightarrow \text{NO}_3^-$) reduces nitrogen loss through denitrification ($\text{NO}_2^- \rightarrow \text{NO}$) [67].
However, the source of NO_2^- in the MT sediments is unknown because the NO_2^-
concentration was below the detection limit ($<0.1 \mu\text{M}$) [10]. A recent study showed
that the microbial nitrate reduction process was stimulated in the MT sediments [68],
suggesting that NO_2^- is supplied to the sediment through the microbial nitrate
reduction process or from the environment [58, 69].

Seven NOB genomes were recovered from the metagenomes of the MT deep
sediments (Figures 3 and S3). These MAGs were identified as new NOB lineages that
are not classified into the common NOB taxa *Nitrospinota*, *Nitrospirota*,
*Chloroflexota*, or *Proteobacteria*. Notably, each of the seven new NOB was able to
fix inorganic carbon via the rTCA or CBB cycle (Figure 3) [70]. However, the O_2
content in the MT sediment would be much lower than that in the surrounding
seawater, indicating that these hadal NOB had a microaerophilic lifestyle and entered
the hadal sediment through physical conditions or biological processes, similar to the
NOB from *Nitrospira* [58, 71]. As a result, the identification of these new NOBs not
only expands their phylogenetic diversity, but also reveals that nitrite oxidation might
be an important but overlooked microbial carbon fixation process in hadal sediment
[71].

**4.3 Anammox bacteria perform aerobic respiration in the MT sediment**

Identification of the *hzs* gene and recovery of three anammox MAGs in the MT
sediment confirm previous results that anammox is an important nitrogen removal
process in hadal sediment environments [6, 24, 61]. Notably, the anammox bacterium
(D.200.bin4.133) was able to perform aerobic respiration because it contained a gene
encoding type A cytochrome *c* oxidase (Figure S7). There are three types of
cytochrome *c* oxidases (types A, B, and C), of which type A cytochrome *c* oxidase

mainly functions in high oxygen concentration environments compared to the other
two types [72]. Type A cytochrome *c* oxidase is also the terminal oxidase with the
highest efficiency in energy generation [73]. In some bacterial strains containing type
A and type C cytochrome *c* oxidases (e.g. *Pseudomonas aeruginosa* PAO1 and
*Shewanella oneidensis* MR-1), type A is highly expressed, while type C remains
unchanged under nutrient-limiting conditions [74, 75]. This implies that type A
cytochrome *c* oxidase is better adapted to oligotrophic environments such as the MT
deep sediment compared to type C cytochrome *c* oxidase. In addition, a phylogenetic
tree containing a collection of 229 type A cytochrome *c* oxidase reference genes
revealed that D.200.bin4.133 might have acquired the type A cytochrome *c* oxidase
gene by horizontal gene transfer from *Desulfobacterota* (Figure S10).

Furthermore, reconstruction of the metabolic pathway of the anammox bacterium
(D.200.bin4.133) showed that this strain can obtain energy through the utilization of
several carbohydrates (glucose, lactose, starch/glycogen, or oligosaccharides) (Figure
4), which would provide more ATPs than anaerobic metabolism when O₂ is the
terminal electron acceptor [76, 77]. Anammox bacteria have demonstrated their ability
to live in oxygenated environments. For example, “*Candidatus Brocadia caroliniensis*”
and “*Candidatus Kuenenia stuttgartiensis*” were observed to tolerate oxygen
concentrations of up to 120 μM and 200 μM, respectively [78]. The defense
mechanisms against oxygen in these anammox bacteria may be attributed to the
expression of enzymes such as bilirubin oxidase, cytochrome *c* oxidase and
bifunctional catalase-peroxidase [79]. Similarly, the gene encoding type A cytochrome
*c* oxidase in the genome of D.200.bin4.133 might act as an oxygen scavenger to
maintain the anammox process in an anaerobic environment, which confers a
selective advantage to D.200.bin4.133 that faces O₂ concentration fluctuations in MT
[4, 80, 81]. The O₂ content in the MT seawater below 6,000 mbsl was constantly
within the range between 156 μM and 188 μM [10]. The O₂ source in the MT
sediment may be derived from the overlying seawater intruding through the
seawater-sediment interface [6, 82]. Moreover, the abundant AOA species in the MT

sediment might also produce a small amount of O₂, as recently reported [83, 84], and
the O₂ could be rapidly utilized by other aerobic microorganisms such as anammox
bacterium (D.200.bin4.133). However, further evidence is needed to test the aerobic
respiration activity of MT anammox bacteria, for example through
metatranscriptomics, metaproteomics and rate measurements.

**4.4 Alkane-degrading *Ketobacter* fix N₂ in the MT seawater**

N₂ production through microbial denitrification and anammox processes forms
the largest nitrogen sink in the ocean [24, 85]. Recent studies revealed that most of N₂
is produced via the anammox process in the hadal sediments of the Atacama Trench
(~67%) and the Kermadec Trench (>90%) [6, 24]. In the MT seawater, however, the
denitrification process is responsible for the majority of N₂ production, as nitrate (~36
μM) and denitrifying microorganisms are abundant (Figure 2) [10, 25]. The
accumulated N₂ is then used by an abundant nitrogenase-containing *Ketobacter* strain
436 W.bin6.184 (Figure 5). However, N₂ fixation is an extremely high energy-consuming
process (16 ATPs per mole of N₂ fixed) [86, 87]. Various organic matters could serve
as energy sources for nitrogen fixation, such as cellulose, chitin, glucan, pectin,
polyphenols, starch, and alkane [88, 89]. To meet this large energy requirement,
*Ketobacter* (W.bin6.184) might have the potential to perform aerobic degradation of
medium-chain alkanes to produce acetyl-CoA (>100 ATPs per mole of oxidized
medium-chain alkane) [90] due to the presence of complete gene sets for alkane
oxidation (*alk* and β-oxidation) (Figure 5). This is consistent with the features of most
genera of *Alcanivoracaceae*, a well-known aerobic hydrocarbon-degrading bacterial
family [91, 92]. Of note, the *alkB* gene was detected in most of the *Ketobacter*
genomes (Table S8), and the utilization of n-alkane by a pure *Ketobacter* strain was
demonstrated by incubation experiments [91], suggesting that *Ketobacter* W.bin6.184
could obtain energy from alkane degradation. The concentration of n-alkanes was
23.5 μg/gdw in the MT seawater as previously measured [26, 93], and the
alkane-degrading bacteria including the members of *Alcanivoracaceae* were abundant
in the MT seawater (Figure 1). Possible sources of the alkanes in the MT might

include a mixture of biological processes (through rotting organisms) and geological
processes (through water-rock reactions) [94, 95]. The slow degradation of alkanes
may spread over long distances due to the effects of hadal seawater currents [94].

**5. Conclusion**

This study provides new insights into the unique features of microbial nitrogen
cycling processes in the deepest part of the ocean. The distinct dominant microbial
taxa were observed in the different MT habitats. The identification of five new NOB
lineages in the MT sediment has uncovered an overlooked process of inorganic
carbon fixation. Meanwhile, anammox bacteria might perform aerobic respiration in
response to nutrient limitations or O₂ fluctuations in the sediment. In the MT seawater,
an abundant *Ketobacter* strain might obtain energy during alkane degradation, and
then fix N₂ released by sedimentary denitrifiers and anammox bacteria (Figure 6).
Integrating a multi-omics strategy that combines metagenomics with
metatranscriptomics, metaproteomics or metabolomics will provide more
comprehensive understanding of hadal microbial communities in future studies.
Meanwhile, laboratory incubation experiments and *in situ* activity tests are also
needed to verify the contribution of hadal microbial communities to global
biogeochemical nitrogen cycles.

**Author's contribution statement**

**Yuhan Huang:** Conceptualization, Formal analysis, Visualization,
Writing-Original Draft. **Xinxu Zhang:** Conceptualization, Methodology, Software,
Resources, Investigation, Writing-Original Draft. **Yu Xin:** Methodology,
Writing-Review & Editing. **Jiwei Tian:** Methodology, Writing-Review & Editing.
**Meng Li:** Supervision, Writing-Review & Editing, Funding acquisition.

**Declarations of Competing Interest**

The authors declare that they have no known competing financial interests or
personal relationships that could have appeared to influence the work reported in this

paper.

**Data Availability**

The genome sequences from the current study have been deposited in eLMSG (an
eLibrary of Microbial Systematics and Genomics, <https://www.biosino.org/elmsg/>)
under accession numbers (MAGs) LMSG_G000011485.1-LMSG_G000011652.1.
Raw reads of the metagenomes for MT deep sediments have been deposited in NODE
(The National Omics Data Encyclopedia, <https://www.biosino.org/node/>) under the
accession number OEP003780.

**Funding**

This work was supported by the National Natural Science Foundation of China
(Grant Nos. 92251306, 91951102, 32070108, 32225003), Guangdong Major Project
of Basic and Applied Basic Research (2023B0303000017), the Southern Marine
Science and Engineering Guangdong Laboratory (Zhuhai) (SML2023SP218),
Shenzhen University 2035 Program for Excellent Research (2022B002) and the
Synthetic Biology Research Center of Shenzhen University.

**Acknowledgements**

We thank Wan Liu for the genome submission to eLMSG
(<https://www.biosino.org/elmsg/index>).

**References**

[revised manuscript text omitted]

35. Zhang M, Zhang XX, Tran NT, Sun ZQ, Zhang XS, Ye HH, Zhang YL, Ma
HY, Aweya JJ, Li SK. 2021. Molting alters the microbiome, immune response,
and digestive enzyme activity in mud crab (*Scylla paramamosain*). *mSystems*
6:e0091721. <https://doi.org/10.1128/mSystems>.

36. Hiraoka S, Hirai M, Matsui Y, Makabe A, Minegishi H, Tsuda M, Juliarni,
Rastelli E, Danovaro R, Corinaldesi C, Kitahashi T, Tasumi E, Nishizawa M,
Takai K, Nomaki H, Nunoura T. 2020. Microbial community and geochemical
analyses of trans-trench sediments for understanding the roles of hadal
environments. *ISEM J* 14:740-756.
<https://doi.org/10.1038/s41396-019-0564-z>.

37. NCBI Resource Coordinators. 2016. Database resources of the National
Center for Biotechnology Information. *Nucleic Acids Res* 44:D7-D19.
<https://doi.org/10.1093/nar/gkv1290>.

38. Nayfach S, Roux S, Seshadri R, Udwy D, Varghese N, Schulz F, Wu DY,
Paez-Espino D, Chen IM, Huntemann M, Palaniappan K, Ladau J, Mukherjee
S, Reddy TBK, Nielsen T, Kirton E, Faria JP, Edirisinghe JN, Henry CS,
Jungbluth SP, Chivian D, Dehal P, Wood-Charlson EM, Arkin AP, Tringe SG,
Visel A, Woyke T, Mouncey NJ, Ivanova NN, Kyrpides NC, Elie-Fadrosh EA,

[revised manuscript text omitted]

909

910